# Dietary Intake and Circulating Amino Acid Concentrations in Relation with Bone Metabolism Markers in Children Following Vegetarian and Omnivorous Diets

**DOI:** 10.3390/nu15061376

**Published:** 2023-03-12

**Authors:** Jadwiga Ambroszkiewicz, Joanna Gajewska, Joanna Mazur, Katarzyna Kuśmierska, Witold Klemarczyk, Grażyna Rowicka, Małgorzata Strucińska, Magdalena Chełchowska

**Affiliations:** 1Department of Screening Tests and Metabolic Diagnostics, Institute of Mother and Child, Kasprzaka 17a, 01-211 Warsaw, Poland; 2Department of Humanization in Medicine and Sexology, Collegium Medicum, University of Zielona Gora, 65-729 Zielona Gora, Poland; 3Department of Nutrition, Institute of Mother and Child, Kasprzaka 17a, 01-211 Warsaw, Poland

**Keywords:** vegetarian diet, protein, amino acids, bone metabolism markers, albumin, prealbumin, children

## Abstract

Scientific studies reported that most vegetarians meet the total protein requirements; however, little is known about their amino acid intakes. We aimed to assess dietary intake and serum amino acid levels in relation to bone metabolism markers in prepubertal children on vegetarian and traditional diets. Data from 51 vegetarian and 25 omnivorous children aged 4–9 years were analyzed. Dietary intake of macro- and micronutrients were assessed using the nutritional program Dieta 5^®^. Serum amino acid analysis was performed using high-pressure liquid chromatography technique, 25-hydroxyvitamin D and parathormone–electrochemiluminescent immunoassay, and bone metabolism markers, albumin, and prealbumin levels using enzyme-linked immunosorbent assay. Vegetarian children had a significantly lower intake of protein and amino acids with median differences of about 30–50% compared to omnivores. Concentrations of four amino acids (valine, lysine, leucine, isoleucine) in serum varied significantly by diet groups and were lower by 10–15% in vegetarians than meat-eaters. Vegetarian children also had lower (*p* < 0.001) serum albumin levels compared to omnivores. Among bone markers, they had higher (*p* < 0.05) levels of C-terminal telopeptide of collagen type I (CTX-I) than omnivores. Correlation patterns between amino acids and bone metabolism markers differed in the vegetarian and omnivore groups. Out of bone markers, especially osteoprotegerin was positively correlated with several amino acids, such as tryptophan, alanine, aspartate, glutamine, and serine, and ornithine in vegetarians. Vegetarian children consumed apparently sufficient but lower protein and amino acids compared to omnivores. In circulation these differences were less marked than in the diet. Significantly lower amino acid intake and serum levels of valine, lysine, leucine, and isoleucine as well as the observed correlations between serum amino acids and biochemical bone marker levels indicated the relations between diet, protein quality, and bone metabolism.

## 1. Introduction

Nutritional habits have been considered an important modifiable factor affecting bone health. The organic matrix of bone consists of collagen and a variety of non-collagenous proteins, so adequate dietary intake of protein, which plays structural, kinetic, catalytic, and signaling roles, seems to be essential for optimal acquisition and maintenance of bone mass [1,2]. Special attention has been focused on a balanced diet and adequate protein intake in childhood and adolescence periods of intensive growth and development. Children require more energy and nutrients per body weight unit compared to adults to obtain normal development of the endocrine, neural, and immunological systems [3,4,5]. They need not only adequate dietary protein intake but also adequate amounts of amino acids (AA), including essential amino acids (EAA), which cannot be synthesized endogenously.

Amino acids may be essential precursors for the synthesis of many important molecules and regulate key metabolic pathways and processes that are significant to the health, growth, and homeostasis of organisms [6]. Thus, an optimal balance among amino acids in the diet and in the circulation deserves special attention. Recent data demonstrated that AAs should also be viewed as specific and selective signaling molecules in bone cells [7,8]. Bone mass can be elevated by amino acid-induced increases in calcium absorption efficiency, osteoblast proliferation and bone mineralization, synthesis of type I collagen, circulating levels of insulin-like growth factor-I (IGF-I) and alkaline phosphatase, suppressed osteoclast differentiation and reduced bone resorption [9,10]. Certain AA types, particularly amino acids from the aromatic group (phenylalanine, tryptophan, tyrosine) can stimulate an increase in intracellular calcium and extracellular signal-regulated kinase (ERK) phosphorylation/activation [8,11]. The branched-chain amino acids (BCAA) such as valine, leucine, and isoleucine are the most potent stimulators of muscle protein synthesis, which is critical also for maintaining adequate bone strength and density [12,13].

Over the last years, vegetarian diets which exclude the consumption of meat, have been gaining popularity in industrialized countries, including families with young children [14]. Additionally, a growing number of children have been born to vegetarian mothers [15]. Although it is accepted that appropriately planned vegetarian diets are nutritionally adequate for individuals during all stages of the life cycle [16,17], concerns exist about their potential insufficiency in regard to some nutrients, especially in vegans.

Several observational studies provided evidence for health benefits from a plant-based diet, including reduced risk of chronic diseases; however, the majority of research has been conducted on adult’ populations and little is known about the possible health consequences of vegetarianism in infants and children [18,19,20,21,22,23,24,25]. Vegetarian diets may be connected with lesser bioavailability and insufficiency of nutrients such as protein (amino acid composition), minerals (calcium, iron, zinc), and vitamins (vitamin B12, vitamin D), which play a critical role in maintaining muscle mass and bone health [1,26]. Inadequate nutrient composition, especially at a younger age, could contribute to bone remodeling imbalance, failure to achieve optimal bone mass, and higher risk of osteoporosis in later life.

There are reports suggesting that long-term vegetarian diets, especially a restricted vegan diet, are associated with lower bone mineral density (BMD) and higher risk of bone fractures compared to those on an omnivorous diet [27,28,29]. Apart from measuring BMD, biochemical bone metabolism markers such as osteocalcin (OC), cross-linking telopeptide of collagen type I (CTX-I), osteoprotegerin (OPG), IGF-I, parathormone (PTH), and 25-hydroxyvitamin D are clinically useful in the assessment of bone turnover [30,31]. To assess the balance between the processes of bone formation and resorption, the OC/CTX-I ratio can be use. Subjects following vegetarian diets seem to have increased parathormone levels and bone resorption markers [32]. Whether potential differences in amino acid profile caused by consuming vegetarian diets can affect bone status in children and adolescents has not yet been investigated.

Therefore, the aims of the present study are to (i) compare dietary intake and serum concentrations of amino acids in prepubertal children following vegetarian and omnivorous diets, (ii) investigate serum levels of bone metabolism markers in both studied groups, and (iii) assess relationships within serum amino acids and bone marker concentrations in the examined groups of children.

## 2. Materials and Methods

### 2.1. Subjects

In this cross-sectional study, we recruited 76 prepubertal children (age range 4–9 years) between June 2020 and December 2021 from a group of consecutive patients attending the Department of Nutrition at the Institute of Mother and Child in Warsaw (Poland). All studied children were Caucasians. Among them, 51 children were following vegetarian diets, free of meat products, with 82.7% of them described as lacto-ovo-vegetarians (include dairy and eggs); 7.7% of them lacto-vegetarians (consume dairy but not eggs) and 9.6% with more restrictive dietary pattern such as vegans (exclude both dairy and eggs). The examined children remained under regular medical and nutritionist care. We recruited the maximum possible number of prepubertal children consuming a vegetarian diet since birth. The vegan children were breastfed by vegan mothers.

The inclusion criteria regarding vegetarians were: being on a diet excluding meat from birth, having no signs of puberty, generally healthy (without development and nutrition disorders), normal-weight with BMI z-score between −1 and +1 [33]. The exclusion criteria were: not in the prepubertal period, history of low birth weight, gastrointestinal diseases accompanied by malabsorption, history of chronic infection, and drug consumption, except for standard vitamin D supplementation. The control group included 25 healthy children following a traditional omnivorous diet (including meat, poultry, fish, dairy, and eggs).

Each participant underwent a basic medical check-up and anthropometric examination using calibrated instruments. Height was measured with a stadiometer and recorded with a precision of 0.5 cm. Weight was assessed unclothed with a calibrated scale to the nearest 0.1 kg. Body mass index (BMI) was calculated as body weight (kg) divided by height squared (m^2^). Based on the data from the questionnaire completed by the parents, the studied children (vegetarians and omnivores) had similar physical activity (PA) and accumulated about 60−90 min/day of moderate-to-vigorous physical activity (MVPA) and approximately 30 min per day of vigorous physical activity (VPA).

The protocol of this study was in accordance with the Helsinki Declaration of Principles and approved by the Ethics Committee of the Institute of Mother and Child (decision number 6/2020, date of approval 6 April 2020). The parents of the participants were comprehensively informed about the study details and signed the informed consent form.

### 2.2. Dietary Assessment

Assessment of dietary intakes in the studied children was done using diet record methodology and had been described in more detail previously [34]. After being advised by a nutritionist, the parents of the studied children prepared a food diary for their children. During the visit, the nutritionist asked for detailed information about the recorded foods and drinks, such as portion sizes and preparation methods using a photo album of products and dishes [35]. When necessary, the food diary was corrected in the presence of the child and parent. The data of the three-day dietary records (two weekdays and one weekend day) were selected and entered into the nutritional software program Dieta 5^®^ (National Food and Nutrition Institute, Warsaw, Poland) [36]. The obtained data were compared with the current age- and sex-specific dietary recommendations for Polish children [37]. The average daily dietary energy, protein, fiber, calcium, phosphorus, magnesium, vitamin D, and amino acid intakes were assessed in 61 (80%) of the studied children: 41 (80.4%) of vegetarians and 20 (80%) of omnivores. Due to the assay method used, data of the intake of 18 amino acids (expressed in mg per day) were estimated.

### 2.3. Biochemical Analyses

For biochemical measurements, venous blood samples (3.5 mL) were collected in the morning hours after an overnight fast to avoid short-term dietary influence and diurnal variations. The samples were centrifuged at 2500× *g* for 10 min at 4 °C and serum was obtained. Serum samples were aliquoted at 100 or 200 μL portions into Eppendorf tubes and stored at −80 °C no longer than two months until assay. Biochemical parameters were assessed in all the studied children, except for PTH concentration, which was determined in 61 (80.3%) subjects. Serum concentrations of 25-hydroxyvitamin D and PTH were determined by electrochemiluminescent immunoassay (ECLIA) using kits from DiaSorin Inc. (Stillwater, OK, USA) on a Liaison analyzer with a precision of the coefficient of variation (CV) of 6.0–9.8%.

Serum amino acid analysis was performed by high-pressure liquid chromatography reversed-phase separation and fluorescence detection (Ex-340 nm and Em-460 nm) using the HPLC-RF-10AXL system (Shimadzu, Kyoto, Japan). Primary amino acids were derivatized with o-phthalaldehyde 3-mercaptopropionic acid (OPA) and then with 9-fluorenylmethyl chloroformate (FMOC). Chromatographic separation was achieved by gradient elution (mobile phase 1 with 40 mmol/L phosphate buffer, pH 7.8) and the organic mobile phase 2 with 45% acetonitrile + 45% methanol + 10% water) on column C18 (Gemini 5 µm C18/ODS, 11OA; Phenomenex 4.6 × 150 mm, Torrance, CA, USA).

Serum levels of albumin and prealbumin were assayed using ELISA kits from Bioassay Technology Laboratory (Jiaxing, China), in which intra-assay CVs were below 8% and inter-assay CVs were below 10%. The limit of detection was 0.52 mg/mL for albumin and 2.51 µg/mL for prealbumin. Concentrations of bone metabolism markers (OC, CTX-I, OPG, IGF-I) were assessed using enzyme-linked immunosorbent assay (ELISA), according to the manufacturer’s instructions. Serum levels of OC and CTX-I were detected using N-MID Osteocalcin and Serum CrossLaps (CTX-I) kits from Immunodiagnostic Systems (Boldon, UK). The limit of detection was 0.5 ng/mL for OC, and 0.020 ng/mL for CTX-I. The intra- and interassay CVs were: 1.3–2.2% and 2.7–5.1% for OC and 1.7–3.0% and 2.6–10.9% for CTX-I, respectively. OPG concentrations were assessed using a kit from DRG Instruments GmbH (Marburg, Germany) with a limit of detection of 0.03 pmol/L, intra-assay CV between 2.5 and 4.9%, and inter-assay CV between 1.7 and 9.0%. Serum IGF-I was determined using a kit from Mediagnost (Reutlingen, Germany), where the analytical sensitivity was 0.091 ng/mL, intra-assay CV was between 5.08 and 6.65%, and inter-assay CV was between 5.53 and 6.56%.

### 2.4. Statistical Analyses

The obtained data were statistically analyzed using IBM Statistics for Windows version 27.0 (Amonk, NY, USA, IBM Corp.). All variables were tested for normality using the Kolmogorov–Smirnov test. Data are presented as frequency (percentage), means ± standard deviation (SD) for normally distributed data or medians and interquartile ranges (IQR) for skewed distribution. The ratio of OC to CTX concentrations was calculated. Groups differences were assessed using Student’s *t*-test or the Mann–Whitney U test, as appropriate. Univariate correlation analyses were performed separately in the vegetarian and omnivore groups using the Spearman rank correlation (rho) coefficient. For bone markers showing a significant correlation with more amino acids in the vegetarian group, a multiple linear regression model was estimated using a stepwise method of variable selection. Regression parameter values, R-sq coefficient, and selected covariance statistics for multicollinearity assessment are shown. A tolerance greater than 0.1 or a variance inflation factor (VIF) less than 5 is considered to indicate no covariance. A *p*-value of less than 0.05 was considered to be statistically significant.

## 3. Results

All participants were healthy, normal weight, prepubertal children. Vegetarians and omnivores were comparable in terms of age, sex, and body mass index (Table 1). Analyzing the children’s diets, we observed that both studied groups had similar total energy intake. Vegetarians had a significantly lower percentage of energy from protein (*p* < 0.001), a higher percentage of energy from carbohydrates (*p* < 0.05), and a similar percentage of energy from fat compared with omnivores. Moreover, dietary intake of protein (in grams per day) was significantly lower (*p* < 0.001) in children on a vegetarian diet compared to meat-eaters. As expected, in the vegetarian diet about 63% of the protein comes from plant origin and 37% from animal sources, while in omnivores it was 34% and 66%, respectively. In addition, vegetarians had significantly higher (*p* < 0.01) intake of fiber and lower (*p* < 0.05) intake of calcium and vitamin D. Dietary intakes of phosphorus and magnesium were comparable between the groups.

As shown in Table 2, dietary intake of all amino acids, including essential AAs was significantly lower (*p* < 0.001) in vegetarian children than in omnivores, except for cysteine intake. The largest percentage differences between the diet groups were for intakes of lysine and methionine, which were about 48% and 43% less in vegetarians than in omnivores. 

The amino acid concentrations in the serum of children on vegetarian and omnivorous diets are presented in Table 3. Serum levels of four of the 24 amino acids varied between the diet groups, being significantly lower (*p* < 0.05) in vegetarians than in omnivores. The largest median differences were for essential AAs: isoleucine (about −18%), lysine (about −16%), leucine (about −14%), and valine (about −10%), yet still within the normal ranges according to Blau et al. [38]. Moreover, serum aspartate, glutamate, ornithine, and taurine tended to be lower (by about 8–12%) in children on a vegetarian diet than in omnivores; however, these differences were not significant.

Vegetarian children had lower (*p* < 0.001) serum albumin levels compared to omnivores and comparable prealbumin concentrations (Table 4). Among the bone metabolism markers, serum concentration of bone resorption marker (CTX-I) was significantly higher (*p* < 0.05) in vegetarians than omnivores. Moreover, serum levels of IGF-I and PTH tended to be higher in these children. Other bone marker concentrations, such as osteocalcin, osteoprotegerin, and 25-hydroxyvitamin D were comparable between the diet groups. The similar serum vitamin D levels in our studied groups of children can be explained by the fact that the majority of participants (vegetarians and omnivores) took vitamin D supplements (an average dose of 500 ± 200 IU/day).

Assessing correlations between albumin/prealbumin and bone markers levels, we found that serum albumin concentration negatively correlated with OPG (r = −0.308, *p* = 0.028) and prealbumin with PTH levels (r = −0.533, *p* = 0.006) in vegetarian children. In the omnivorous group, serum prealbumin was correlated with PTH concentration (r = −0.886, *p* = 0.019). Additionally, albumin/prealbumin levels were associated with serum amino acids (data not shown). In vegetarians, albumin negatively correlated with methionine (r = −0.331, *p* = 0.017), tryptophan (r = −0.371, *p* = 0.007), valine (r = −0.330, *p* = 0.018), tyrosine (r = −0.438, *p* = 0.001), alanine (r = −0.348, *p* = 0.012), and proline (r = −0.310, *p* = 0.027). In omnivores, albumin level positively correlated with phenylalanine (r = 0.457, *p* = 0.033) and lysine (r = 0.463, *p* = 0.030). Prealbumin levels did not correlate with amino acids in omnivores, while in vegetarians they inversely correlated with methionine (r = −0.291, *p* = 0.038), leucine (r = −0.297, *p* = 0.036), and glutamine (r = −0.289, *p* = 0.040).

The pattern of correlations between serum amino acids and bone metabolism marker concentrations was different in vegetarians and omnivores (Figure 1).

In vegetarians, we observed that osteocalcin concentrations were significantly negatively correlated with lysine (r = −0,311, *p* = 0.026) and glutamate (r = −0.325, *p* = 0.020), however, positively with glutamine (r = 0.327, *p* = 0.019) levels. CTX-I levels were inversely associated with lysine (r = −0.364, *p* = 0.009) and ornithine (r = −0.369, *p* = 0.004) while directly with tryptophan (r = 0.307, *p* = 0.029) concentrations. We observed positive correlations between osteoproegerin and tryptophan (r = 0.373, *p* = 0.007), alanine (r = 0.294, *p* = 0.042), aspartate (r = 0.314, *p* = 0.025), glutamine (r = 0.287, *p* = 0.041), serine (r = 0.278, *p* = 0.048), and negative with ornithine (r = −0.280, *p* = 0.048) concentrations. Among the other bone markers, only 25-hydroxyvitamin D concentrations were positively correlated with phenylalanine (r = 0.299, *p* = 0.014) levels in this group of children.

In omnivores, serum osteocalcin was directly correlated with tryptophan (r = 0.398, *p* = 0.048) and indirectly with glycine (r = −0.425, *p* = 0.034) levels, while CTX-I was negatively correlated with isoleucine (r = −0.656, *p* = 0.001), leucine (r = −0.543, *p* = 0.006), asparagine (r = −0.418, *p* = 0.038), and proline (r = −0.412, *p* = 0.041) concentrations. OPG was negatively correlated with ornithine (r = −0.424, *p* = 0.034) and taurine (r = −0.405, *p* = 0.045), while IGF-I with cysteine (r = −0.393, *p* = 0.050). The levels of 25-hydroxyvitamin D were directly correlated with several amino acids such as phenylalanine (r = 0.450, *p* = 0.024), threonine (r = 0.579, *p* = 0.002), leucine (r = 0.374, *p* = 0.050), lysine (r = 0.473, *p* = 0.017), arginine (r = 0.452, *p* = 0.023), asparagine (r = 0.476, *p* = 0.016), glutamate (r = 0.439, *p* = 0.028), serine (r = 0.664, *p* = 0.001), and tyrosine (r = 0.542, *p* = 0.005). However, PTH concentrations were negatively related with methionine (r = −0.891, *p* = 0.001), phenylalanine (r = −0.901, *p* = 0.001), isoleucine (r = −0.821, *p* = 0.023), leucine (r = −0.964, *p* = 0.001), lysine (r = −0.786, *p* = 0.036), asparagine (r = −0.919, *p* = 0.003), serine (r = −0.786, *p* = 0.036), tyrosine (r = −0.857, *p* = 0.014), and taurine (r = −0.750, *p* = 0.043) levels.

Table 5 shows the results of multiple linear regression estimation with osteoprotegerin as the dependent variable. Six significantly correlating amino acids (tryptophan, alanine, aspartate, glutamine, serine, ornithine) and albumin were included as independent variables. Three factors entered the final model, in order: alanine, ornithine and aspartate. The model explains 35.1% of the variation in osteoprotegerin in the vegetarian group, and the tolerance and VIF indices are highly favorable.

## 4. Discussion

The adequacy of dietary protein intake from vegetarian diets has been discussed in the literature. Because the overall protein quality of a vegetarian diet is estimated to be about 80–90% compared with the meat-eater diet (mainly due to the lower digestibility of plant protein), it has been suggested that the dietary requirements of vegetarians should be increased by about 20% [26,39]. Most studies were conducted on adult vegetarians [40,41,42,43]. Despite lower average protein intake in the plant-based dietary patterns, researchers reported that vegetarians had protein intakes still within the recommended range [44,45]. The present study showed that children following a vegetarian diet had lower but sufficient dietary intake of protein. They also had lower dietary intake of amino acids with the highest median differences in essential AAs. Despite this, the serum levels of amino acids were within the reference values, but significantly lower in the case of lysine, leucine, isoleucine, and valine than in omnivores.

Adequate protein intake during development is critical to ensure optimal peak bone mass in later life. Insufficient protein intake leads to growth retardation during early life stages and poor bone quality in adults. The effect of protein on bone metabolism is related to the capacity to provide essential AAs for the synthesis of the bone collagen matrix. Protein restriction is also associated with a decrease in the synthesis of IGF-I, an anabolic factor for bone and muscle. In an animal model, Rouy et al. [46] described lower bone mineral density and reduced bone turnover markers in animals fed a soy-based protein-restricted diet compared to a diet containing normal amount of protein as well as with a casein-based protein-restricted diet. This can be partly related to a difference in AA profile, as casein is richer than soy in methionine, proline, serine, threonine, glutamine, valine, tyrosine, isoleucine, and leucine with a possible effect on bone protein turnover. Thus, protein quality appears critical to ensure adequate growth and optimal peak bone mass. Moreover, animals subjected to a methionine restricted diet had significantly lower BMD and bone mineral content (BMC), possibly due to decreased IGF-I levels leading to a disrupted IGF-I/IGFBP axis [47].

Only a few studies analyzed the amino acid profile in vegetarians, mostly in adults [48,49,50]. The authors found that dietary intakes of AAs were the lowest in vegans followed by vegetarians, then fish-eaters, and the highest in meat-eaters. The largest differences were found for lysine and methionine intakes and plasma levels of essential amino acids between diet groups [48,49]. In vegan children, Hovinen et al. [51] reported lower protein intake and plasma concentrations of leucine, isoleucine, lysine, phenylalanine, and valine. Generally, the dietary profile of AA was less optimal in plant foods than in animal foods. Plant-based proteins contain lower amounts of certain essential amino acids (e.g., leucine, lysine, methionine) and higher amounts of arginine, glutamine, and glycine compared with animal-based proteins [1,52].

Existing data demonstrated a direct effect of AAs on bone metabolism through complex cellular pathways. Fini et al. [53] conducted a study on primary osteoblast cultures from normal and osteopenic rats. They reported the effects of lysine and arginine on the stimulation of intestinal calcium absorption and the cross-linking process of bone collagen, which is essential for bone matrix formation. Arginine is also involved in the synthesis of IGF-I and proline, which acts as substrate for collagen production. Treatment with lysine and arginine resulted in an increased level of bone formation markers such as procollagen I *C*-terminal propeptide (PICP) and alkaline phosphatase. In turn, the combination of phenylalanine, tyrosine, and tryptophan treatment may lead to reduced osteoclast differentiation, and result in a lower rate of resorption [54].

Serum albumin and prealbumin (also called transthyretin) are widely used biomarkers in assessing nutritional status [55]. Caso et al. [56] observed a reduced rate of albumin synthesis in adult vegetarians compared with subjects consuming a traditional diet, suggesting that different food sources might have a different effect on albumin synthesis. Albumin synthesis might be responsive to a reduction in amino acid availability, a consequence of the lower digestibility, and AA score as well as higher fiber content of vegetarian diets. In our study, vegetarian children compared with omnivores, had significantly lower but still in the normal range serum albumin levels. Additionally, its level was significantly associated with several serum amino acids, such as methionine, tryptophan, valine, tyrosine, alanine, and proline. Lower levels of serum albumin in our examined vegetarian children compared to omnivores might be related to the different proportions of animal/plant protein in the diets [57].

Individuals following a vegetarian diet usually had lower dietary intake with poorer bioavailability not only of protein and essential amino acids but also minerals (e.g., calcium) and vitamins (vitamin D, vitamin B12), which are particularly important for optimal bone health [58]. Research on bone status in subjects on vegetarian diets remains limited. Some reports suggested that vegetarians, especially vegans have increased parathormone levels, higher bone turnover rate, resulting in decreased bone mineral density compared with omnivores [27,28,32,59,60]. Moreover, the results from studies on the association between hypoalbuminemia with osteoporosis are conflicting [61]. Albumin and prealbumin are not only markers of nutrition but are also associated with inflammation. Numerous proinflammatory cytokines have been implicated in the regulation of osteoblasts and osteoclasts playing a potential role in bone remodeling [55,62].

The interpretation of amino acid intakes and serum concentrations in regard to their impact on bone metabolism is difficult because of the limited research in this field and a relative lack of reference values of these parameters for children. In the present study, vegetarian children had higher serum levels of CTX-I and comparable concentrations of osteocalcin and the OC/CTX-I ratio, suggesting no significant difference in bone turnover rate. Analyzing the correlations, we observed that several amino acids were associated with bone metabolism markers, but their patterns were different in vegetarian and omnivorous children. An interesting finding is that in vegetarian children we noted positive correlations between OPG with several amino acids. A multiple linear regression analysis confirmed that osteoprotegerin is significantly associated with alanine, ornithine, and aspartate. OPG plays a role in the regulation of bone formation/resorption processes through the RANKL/RANK/OPG pathway. It acts as a decoy receptor, which after binding to the receptor activator of nuclear factor kappa B (RANK), blocks the possibility of binding its ligand (RANKL) and consequently decreases bone resorption [63]. An elevated ratio of RANKL/OPG is associated with an imbalance in osteoclastic/osteoblastic activity, leading to increased bone resorption and decreased bone mass.

Recent data demonstrated that amino acids should be viewed as specific and selective signaling molecules in bone cells [59]. Specific AAs may preferentially bind to a calcium-sensing receptor, modulate its activation by calcium ions, and thus potentially impact bone turnover. Amino acids are also required for bone marrow stromal cells to promote differentiation into osteoblasts, synthesis of type I collagen, circulating levels of IGF-I and alkaline phosphatase [9,10]. Certain AA types, particularly amino acids from the aromatic group, can stimulate increases in intracellular calcium and ERK phosphorylation/activation and may impact an early stage of osteoclast differentiation by suppressing the attachment of osteoclasts and the resorption process. The branched-chain AAs, are the most potent stimulators of muscle protein synthesis, which is critical for the maintenance of adequate muscle mass, bone strength, and bone density. The dietary intakes as well as circulating levels of these amino acids were indeed lower in the vegetarian children we examined compared with omnivores. Leucine has a direct effect on the regulation of protein turnover through the mTOR (mammalian target of rapamycin) kinase signaling pathway, valine and isoleucine maintain a balance among BCAAs, while lysine acts among others via the regulation of nitric oxide synthesis [6,64]. It seems that different subclasses of amino acids have varying effects in the body, thus, paying attention to amino acid composition may be relevant to the prevention of bone abnormalities.

The skeleton is a dynamic and metabolically active tissue, and is exquisitely sensitive to its microenvironment [28,57,60]. For example, glutamate and aspartate might be higher (20–70-fold) in the bone microenvironment than in the circulation. Long-term follow-up studies are needed to clarify the consequences of lower protein, essential amino acids, calcium, vitamin D, and albumin in children on a vegetarian diet on bone health.

We realize that our study has several limitations that may impact our interpretation. First, the cross-sectional design does not allow to establish causal relationships. However, this is the first observational study performing the evaluation of dietary and serum amino acid levels in relation with bone metabolism markers in prepubertal children consuming vegetarian and omnivorous diets. Second, our study was limited to prepubertal children who had their nutritional and biochemical data measured at the same time. Thus, our sample size was relatively small, especially for the group of control children. The paper only gives an example of multivariate analysis, as the small sample size is a limitation here. However, the sample was homogenous and unique. All participants were of Polish origin comparable in terms of age, sex, and development stage. Large groups of healthy children are rarely studied because of ethical considerations regarding blood sampling in healthy children. Third, we used three-day (two weekdays and one weekend day) food records to assess dietary intake of macro- and micronutrients. Although not ideal, this method is the most practical and commonly used. In our study, about 20% of the studied children had missing data in the dietary assessment. Fourth, we did not measure bone mineral density and our results regarding bone markers were based on single measurements. However, we assessed bone formation as well as bone resorption markers and could calculate the OC/CTX-I ratio, assessing the rate of bone turnover. In a future study, we are planning to assess the novel bone markers such as RANKL, sclerostin, Dickkopf-related protein-1 (Dkk-1). There are several advantages to using a wide panel of bone markers involved in the regulation of bone turnover through signaling pathways. Since significant correlations between AAs and OPG concentrations were observed in the examined vegetarians, it is worth evaluating the regulatory RANK/RANKL/OPG system. Moreover, the Wnt-beta-catenin signaling pathway, with sclerostin and Dickkopf-related protein-1 (Dkk-1), is crucial in bone metabolism.

To summarize, we observed that children on vegetarian diets consumed apparently sufficient but lower protein and amino acids than omnivores. However, in circulation, the amino acid concentrations were less marked than in the diet. Despite vegetarians having higher levels of bone resorption marker CTX-I, they had a similar concentration of osteocalcin and the OC/CTX-I ratio compared to omnivores. The observed significant associations between serum levels of bone metabolism markers and amino acid concentrations confirm an existing link between protein quality and bone turnover. Particularly, significant relationships of osteoprotegerin with alanine, ornithine, and aspartate might suggest an impact of diet on the bone regulatory pathway. Further investigations could be important to confirm the effect of an altered amino acid profile on bone metabolism in vegetarian subjects and to fully understand the mechanisms involved in the association between nutrition and bone status.

## Figures and Tables

**Figure 1 nutrients-15-01376-f001:**
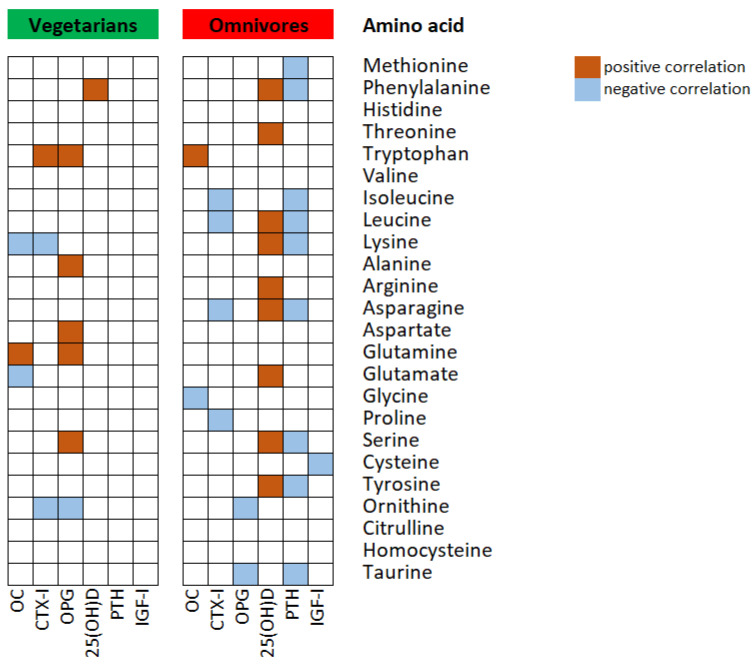
Spearman correlation between serum concentrations of bone metabolism markers and amino acids in children following vegetarian and omnivorous diets. Data of 51 vegetarians and 25 omnivores were analyzed and only statistically significant correlations with *p* < 0.05 are shown. OC—osteocalcin; CTX-I—*C*-terminal telopeptide of collagen type I; OPG—osteoprotegerin; 25(OH)D—25-hydroxyvitamin D; PTH—parathormone; IGF-I—insulin-like growth factor-I.

**Table 1 nutrients-15-01376-t001:** Characteristics and dietary intake of energy, macro-, and micronutrients in vegetarian and omnivorous children.

	Vegetarians	Omnivores	*p* Value
Characteristics			
n	51	25	
Girls, n (%)	25 (49%)	12 (48%)	
Age (years)	6.0 (5.0–8.5)	5.5 (4.5–7.5)	0.1027
BMI (kg/m^2^)	14.9 ± 0.8	15.1 ± 1.2	0.5731
Dietary intake			
n	41	20	
Energy (kcal/d)	1396 (1068–1662)	1476 (1285–1698)	0.4082
Protein, % of energy	12.8 ± 1.8	15.8 ± 3.0	0.0002
Fat, % of energy	30.5 ± 5.0	32.0 ± 3.3	0.3792
Carbohydrates, % of energy	56.7 ± 5.9	52.2 ± 3.8	0.0125
Protein (g/d)	35.5 (28.8–47.7)	54.9 (44.2–66.1)	0.0009
Animal protein (g/d)	13.0 (7.5–17.5)	36.8 (26.7–46.0)	0.0001
Plant protein (g/d)	22.1 (18.1–26.5)	18.7 (14.2–21.7)	0.0106
Fiber (g/d	16.7 (12.4–21.3)	14.8 (11.1–16.9)	0.0034
Calcium (mg/d)	504.7 (311.7–552.5)	616.5 (429.2–755.0)	0.0149
Phosphorus (mg/d)	774.4 (552.1–938.6)	897.6 (708.0–1038.5)	0.1403
Magnesium (mg/d)	229.8 ± 96.5	217.4 ± 95.4	0.6609
Vitamin D (µg/d)	1.70 (0.97–5.30)	2.30 (1.28–5.66)	0.0116

Data are reported as percentage (%); mean ± standard deviation (SD) for normally distributed variables; median and interquartile ranges (IQR) for skewed variables; BMI–body mass index.

**Table 2 nutrients-15-01376-t002:** Dietary intake of amino acids in children on vegetarian and omnivorous diets.

Amino Acids (mg/d)	Vegetarians (*n* = 41)	Omnivores (*n* = 20)	Median Difference (%)	*p* Value
Essential amino acids				
Methionine	750 (623–921)	1322 (1001–1622)	−43.3	0.0002
Phenylalanine	1666 (1301–2112)	2431 (1956–2981)	−31.5	0.0021
Histidine	889 (712–1146)	1486 (1123–1929)	−39.5	0.0007
Threonine	1341 (1054–1743)	2254 (1771–2769)	−40.0	0.0004
Tryptophan	440 (342–574)	670 (552–808)	−34.3	0.0017
Valine	2081 (1587–2569)	3150 (2605–3754)	−33.9	0.0008
Isoleucine	1674 (1292–2055)	2670 (2096–3233)	−37.3	0.0005
Leucine	2743 (2056–3363)	4274 (3456–5331)	−35.8	0.0005
Lysine	1963 (1477–2368)	3762 (2867–4728)	−47.8	0.0001
Non-essential amino acids				
Alanine	1638 (1266–2018)	2632 (1886–3162)	−37.8	0.0006
Arginine	1948 (1541–2529)	2840 (1974–3151)	−31.4	0.0078
Aspartate	3255 (2578–4110)	4866 (3576–5388)	−33.1	0.0001
Glutamate	7346 (5770–9629)	11,157 (8677–14,037)	−34.2	0.0026
Glycine	1400 (1100–1763)	2130 (1520–2410)	−34.3	0.0033
Proline	2794 (1909–3257)	4166 (3413–5063)	−32.9	0.0007
Serine	1807 (1464–2350)	2703 (2199–3117)	−33.1	0.0028
Cysteine	658 (502–855)	778 (549–961)	−15.4	0.1273
Tyrosine	1245 (966–1527)	2041 (1648–2422)	−39.0	0.0006

Data are reported as median values and interquartile ranges (IQR).

**Table 3 nutrients-15-01376-t003:** Concentrations of amino acids in the serum of children on vegetarian and omnivorous diets.

Amino Acids (µmol/L)	Vegetarian Children (*n* = 51)	Omnivorous Children (*n* = 25)	Median Difference (%)	*p* Value
Essential amino acids				
Methionine	20.5 (17.9–24.7)	19.2 (17.0–24.8)	+6.3	0.7699
Phenylalanine	55.9 (45.7–64.5)	54.1 (50.3–65.5)	+3.3	0.7574
Histidine	72.7 (65.2–84.1)	76.5 (68.5–82.2)	−5.0	0.5363
Threonine	97.4 (93.8–118.7)	100.8 (84.7–122.8)	−3.4	0.7099
Tryptophan	29.1 (21.4–36.6)	29.9 (22.1–35.5)	−2.7	0.7139
Valine	183.8 (166.6–208.2)	204.5 (186.3–244.7)	−10.1	0.0253
Isoleucine	49.7 (43.9–62.0)	60.4 (52.6–73.0)	−17.7	0.0231
Leucine	105.4 (93.3–129.4)	122.6 (104.4–145.7)	−14.1	0.0315
Lysine	118.8 (86.5–146.5)	141.6 (115.6–167.4)	−16.1	0.0297
Non-essential amino acids				
Alanine	327.3 (272.2–391.3)	307.0 (247.1–385.5)	+6.2	0.6520
Arginine	92.7 (81.5–105.9)	94.0 (83.1–107.0)	−1.4	0.9446
Asparagine	58.2 (45.4–69.6)	51.4 (43.9–60.9)	+11.7	0.2397
Aspartate	18.1 (14.4–22.8)	19.6 (16.8–23.5)	−7.7	0.0694
Glutamine	573.7 (537.0–665.0)	550.8 (522.9–612.8)	+4.0	0.0907
Glutamate	53.8 (39.9–71.2)	59.0 (43.3–91.2)	−8.6	0.0985
Glycine	265.7 (215.1–301.9)	242.8 (201.6–263.0)	+8.6	0.1821
Proline	158.6 (128.6–231.9)	144.5 (106.3–198.4)	+8.9	0.2497
Serine	143.6 (130.0–164.2)	138.4 (129.1–160.5)	+5.6	0.8430
Cysteine	147.7 (125.2–163.2)	155.2 (144.3–181.2)	−4.8	0.0840
Tyrosine	57.4 (48.7–68.2)	53.7 (49.5–65.8)	+6.4	0.8247
Ornithine	45.0 (31.4–61.9)	51.2 (42.1–67.2)	−12.1	0.1696
Citrulline	30.3 (25.9–33.6)	28.6 (24.4–32.3)	+5.6	0.2148
Homocysteine	5.9 (5.3–6.9)	6.1 (5.1–7.4)	−3.7	0.9041
Taurine	116.0 (91.7–132.4)	127.5 (110.2–140.9)	−9.0	0.0800

Data are expressed as median values and interquartile ranges (IQR).

**Table 4 nutrients-15-01376-t004:** Serum concentration of biochemical parameters (albumin, prealbumin, and bone metabolism markers) in children on vegetarian and omnivorous diets.

	Vegetarian Children (*n* = 51)	Omnivorous Children (*n* = 25)	*p* Value
Albumin (mg/mL)	51.5 (45.0–56.9)	62.9 (57.3–68.1)	0.0001
Prealbumin (µg/mL)	244.1 (215.3–289.6)	255.4 (243.7–277.1)	0.6920
25-hydroxyvitamin D (ng/mL)	29.9 (25.6–37.7)	30.9 (26.7–37.0)	0.8691
PTH (pg/mL)	22.3 (16.4–33.4)	17.4 (10.8–23.9)	0.1733
IGF-I (ng/mL)	159.6 (115.6–225.9)	134.8 (103.6–19.3)	0.0654
OC (ng/mL)	74.8 (55.9–104.1)	75.3 (64.2–89.7)	0.5555
CTX-I (ng/mL)	1.917 (1.541–2.209)	1.711 (1.436–1.928)	0.0343
OC/CTX-I	0.39 (0.31–0.50)	0.42 (0.37–0.64)	0.0915
OPG (pmol/L)	4.0 (3.4–4.9)	4.3 (3.8–5.1)	0.5455

Data are reported as median and interquartile ranges (IQR); PTH–parathormone; IGF-I–insulin-like growth factor-I; OC–osteocalcin; CTX-I–*C*-terminal telopeptide of collagen type I; OPG–osteoprotegerin.

**Table 5 nutrients-15-01376-t005:** Estimation of the final multiple linear regression for OPG in vegetarians (*n* = 51).

	Unstandardized Coefficient		Standardization Factor	t	*p* Value	Covariance Statistics	
	B	Standard Error	β			Tolerance	VIF
Constant	2.742	0.582		4.71	0.000		
Alanine	0.004	0.001	0.366	3.00	0.004	0.927	1.078
Ornithine	−0.017	0.005	−0.368	−3.06	0.004	0.953	1.049
Aspartate	0.058	0.024	0.298	2.39	0.021	0.888	1.126

R-sq = 0.351; VIF*—*variance inflation factor.

## Data Availability

The data analyzed during the current study are available from the corresponding author on reasonable request.

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
