# Peer review of "Dietary Intake and Circulating Amino Acid Concentrations in Relation with Bone Metabolism Markers in Children Following Vegetarian and Omnivorous Diets"

_nutrients, 2023, doi:10.3390/nu15061376_

Round 1

Reviewer 1 Report

The manuscript is well written, and the study is interesting, the obtained data are original and the scientific question is important, as the data on the metabolic effects of vegetarian diets in children are rare. General remark : the first part of the discussion includes mainly the repetition of the obtained results, without really « discussing » what they represent for the nutritional status of the children, do they meet the diet requirements for individual AA intake and how they are related to the metabolism. The second part of the discussion on the relationship between the plasma AA concentrations and bone metabolism markers is very light. It is difficult to understand what the authors mean, there are  only slight attemps to relate AA to bone metabolism markers, that are not very clear. It is possible that the lack of information does ot permit to expand the discussion, but in this case I suggest to do the discussion more short, more concised and concentrated on the possible relation to bone metabolism, instead of rewriting the results and of the the data obtained in other studies that have only slight connexion with the present work.

Results :

Table 2 : Please correct the typo threonine (and not treonine)

Discussion :

General : I suggest to include the following studies into discussion :

Fayth L Miles, Jan Irene C Lloren, Ella Haddad, Karen Jaceldo-Siegl, Synnove Knutsen, Joan Sabate, Gary E Fraser, Plasma, Urine, and Adipose Tissue Biomarkers of Dietary Intake Differ Between Vegetarian and Non-Vegetarian Diet Groups in the Adventist Health Study-2, The Journal of Nutrition, Volume 149, Issue 4, April 2019, Pages 667–675

Miles, F.L.; Orlich, M.J.; Mashchak, A.; Chandler, P.D.; Lampe, J.W.; Duerksen-Hughes, P.; Fraser, G.E. The Biology of Veganism: Plasma Metabolomics Analysis Reveals Distinct Profiles of Vegans and Non-Vegetarians in the Adventist Health Study-2 Cohort. Nutrients 2022, 14, 709. https://doi.org/10.3390/nu14030709

Hernández-Alonso, P., Becerra-Tomás, N., Papandreou, C., Bulló, M., Guasch-Ferré, M., Toledo, E., Ruiz-Canela, M., Clish, C. B., Corella, D., Dennis, C., Deik, A., Wang, D. D., Razquin, C., Drouin-Chartier, J., Estruch, R., Ros, E., Fitó, M., Arós, F., Fiol, M., Serra-Majem, L., Liang, L., Martínez-González, M. A., Hu, F. B., Salas-Salvadó, J., Plasma Metabolomics Profiles are Associated with the Amount and Source of Protein Intake: A Metabolomics Approach within the PREDIMED Study. Mol. Nutr. Food Res. 2020, 64, 2000178. https://doi.org/10.1002/mnfr.202000178

And the FAO reports (2007, 2013) on dietary protein quality evaluation and recommendations in human nutrition.

Lines 275 to 310 : there is a very long description of several data obtained to date in similar studies, without the connexion to the present study. Please do it more concise.

Lines 311-322 : the paragraph repeats the information that is already given in the Results section. Please do it more concised and discuss, for example, if the dietary intake in vegetarians meets the requirements for children. The lysine and methionine intake evaluation is critical for protein quality assessment, and their bioavailiblity is also critical, whereas the authors do not discuss about this.

Line 330 : the authors write that the AA are implicated in bone biology. Please explain in which aspects of bone biology they are implicated.

Lines 331-334 : what does the balance of asp/asn and glu/gln represent in metabolism ? Why do you provide the ratio of ala to pro , ala to gln and gly to ser ? How do these ratios are related to bone metabolism (or others ?)

Line 335 : I do not understand the phrase that the children were under medical care. What is the relationship with the study ? Or with the health of the children ?

Line 340 : The serum albumin levels in children was lower than in adults observed by Caso et al. How do you explain this ? Is it related to nutritional status of the children ?

Lines 347 to 379 : it is very difficult to understand how the studied bone metabolism markers represent the health of the bones, and how the AA are related to them.

Line 381 : even if there is limited research in the field, there are several studies on bones and protein intake in other models. For example, Rouy E, Vico L, Laroche N, Benoit V, Rousseau B, Blachier F, Tomé D, Blais A. Protein quality affects bone status during moderate protein restriction in growing mice. Bone. 2014  doi: 10.1016/j.bone.2013.10.013., and other works of the authors, that may help to extend the discussion.

Lines 423-424 : the authors write that they assess the bone formation (I did not find the information on bone formation in the results/discussion), as well as OC/CTX ratio is not described. No clear information on bone turnover is presented neither in Results, no Discussion sections.

Line 426 : what are the advantage of other bone markers ? What information will they permit to obtain ? Please make it clear for the reader.

Lines 428-439 : the Conclusion is not clear.

Author Response

Comments and Suggestions for Authors

The manuscript is well written, and the study is interesting, the obtained data are original and the scientific question is important, as the data on the metabolic effects of vegetarian diets in children are rare. General remark : the first part of the discussion includes mainly the repetition of the obtained results, without really « discussing » what they represent for the nutritional status of the children, do they meet the diet requirements for individual AA intake and how they are related to the metabolism. The second part of the discussion on the relationship between the plasma AA concentrations and bone metabolism markers is very light. It is difficult to understand what the authors mean, there are  only slight attemps to relate AA to bone metabolism markers, that are not very clear. It is possible that the lack of information does ot permit to expand the discussion, but in this case I suggest to do the discussion more short, more concised and concentrated on the possible relation to bone metabolism, instead of rewriting the results and of the the data obtained in other studies that have only slight connexion with the present work.

Thank you for taking the time to review our manuscript and providing constructive feedback, which helps to improve the quality of our work. We have carefully considered your comments.  The Discussion section was shortened, and some of its fragments were changed to be more clear. The suggested references were included into manuscript. We hope that these changes have addressed your concerns and improved the clarity and quality of our manuscript.

Results :

Table 2 : Please correct the typo threonine (and not treonine)

We corrected this typo in Table 2.

Discussion :

General : I suggest to include the following studies into discussion :

Fayth L Miles, Jan Irene C Lloren, Ella Haddad, Karen Jaceldo-Siegl, Synnove Knutsen, Joan Sabate, Gary E Fraser, Plasma, Urine, and Adipose Tissue Biomarkers of Dietary Intake Differ Between Vegetarian and Non-Vegetarian Diet Groups in the Adventist Health Study-2, The Journal of Nutrition, Volume 149, Issue 4, April 2019, Pages 667–675

Miles, F.L.; Orlich, M.J.; Mashchak, A.; Chandler, P.D.; Lampe, J.W.; Duerksen-Hughes, P.; Fraser, G.E. The Biology of Veganism: Plasma Metabolomics Analysis Reveals Distinct Profiles of Vegans and Non-Vegetarians in the Adventist Health Study-2 Cohort. Nutrients 2022, 14, 709. https://doi.org/10.3390/nu14030709

Hernández-Alonso, P., Becerra-Tomás, N., Papandreou, C., Bulló, M., Guasch-Ferré, M., Toledo, E., Ruiz-Canela, M., Clish, C. B., Corella, D., Dennis, C., Deik, A., Wang, D. D., Razquin, C., Drouin-Chartier, J., Estruch, R., Ros, E., Fitó, M., Arós, F., Fiol, M., Serra-Majem, L., Liang, L., Martínez-González, M. A., Hu, F. B., Salas-Salvadó, J., Plasma Metabolomics Profiles are Associated with the Amount and Source of Protein Intake: A Metabolomics Approach within the PREDIMED Study. Mol. Nutr. Food Res. 2020, 64, 2000178. https://doi.org/10.1002/mnfr.202000178.

And the FAO reports (2007, 2013) on dietary protein quality evaluation and recommendations in human nutrition.

According to the reviewer`s suggestion, we included proposed references.

 Lines 275 to 310 : there is a very long description of several data obtained to date in similar studies, without the connexion to the present study. Please do it more concise.

We have rewritten this paragraph of the Discussion.

Lines 311-322 : the paragraph repeats the information that is already given in the Results section. Please do it more concised and discuss, for example, if the dietary intake in vegetarians meets the requirements for children. The lysine and methionine intake evaluation is critical for protein quality assessment, and their bioavailiblity is also critical, whereas the authors do not discuss about this.

We changed the Discussion and added short information about the dietary intake of lysine and methionine, which is important for protein quality.

Line 330 : the authors write that the AA are implicated in bone biology. Please explain in which aspects of bone biology they are implicated.

Thank you for this important comment. We changed some fragments of the Discussion to focus more on the possible associations between amino acids and bone metabolism.

Lines 331-334 : what does the balance of asp/asn and glu/gln represent in metabolism ? Why do you provide the ratio of ala to pro , ala to gln and gly to ser ? How do these ratios are related to bone metabolism (or others ?)

According to reliever` suggestions, we shortened our Discussion to be more concentrated on the possible relation between AAs and bone metabolism.

Often, the limitations of an individual amino acid can affect the concentrations of other endogenous AAs. Above mentioned AAs such as asparagine and aspartate as well as glutamine and glutamate are closely dependent on each other: glutamate is reversibly converted to glutamine and aspartate to asparagine. Also,  glycine is reversibly converted to serine, alanine can be synthesized from glutamate, etc. Alanine and proline are closely related to the energy state of the cell in the mitochondria. Therefore, the ratios between AAs tell us more than the concentration of individual AA in the assessment of their metabolic balance. It is important to maintain amino acid balance, especially in case of elimination diet. In our examined children, the ratios of the above mentioned amino acids show that metabolic balance is maintained.

Line 335 : I do not understand the phrase that the children were under medical care. What is the relationship with the study ? Or with the health of the children ?

In the Department of Nutrition of our Institute, there is a team of specialists consisting of doctors and dietitians who provide care for children growing up in families that follow a vegetarian diet. These children often follow vegetarian diets from birth, including a more restrictive vegan diet. The doctors regularly consult these children to assess their development, perform anthropometric measurements, and conduct routine biochemical tests. Meanwhile, the dietitians evaluate the children`s diet and suggest modifications if necessary. Also, children on a traditional diet can come to the Department of Nutrition for evaluation of dietary balance.

Line 340 : The serum albumin levels in children was lower than in adults observed by Caso et al. How do you explain this ? Is it related to nutritional status of the children ?

Caso et al. measured the synthesis rate of albumin as a response to different dietary patterns. They observed that albumin synthesis appears to be modulated by changes in the proportion of animal vs. plant protein in the diet and was lower in vegetarians. Albumin synthesis might be responsive to a reduction in amino acid availability, a consequence of the lower digestibility and AA score as well as higher fiber content of the vegetarian diets. Lower levels of serum albumin in our examined vegetarian children compared to omnivores might be related to the different proportions of animal/plant protein in the diets as well as  the nutritional status of these children. We added it into the Discussion.

Lines 347 to 379 : it is very difficult to understand how the studied bone metabolism markers represent the health of the bones, and how the AA are related to them.

We changed this part of the Discussion to focus more on the relationship between the concentration of  AAs and bone metabolism markers.

Line 381 : even if there is limited research in the field, there are several studies on bones and protein intake in other models.

For example, Rouy E, Vico L, Laroche N, Benoit V, Rousseau B, Blachier F, Tomé D, Blais A. Protein quality affects bone status during moderate protein restriction in growing mice. Bone. 2014  doi: 10.1016/j.bone.2013.10.013., and other works of the authors, that may help to extend the discussion.

Results reported by Rouy et al. regarding the impact of restricted protein diets (soy-based and casein-based) on bone mineral density and bone metabolism markers in animal model were included in the Discussion.

 Lines 423-424 : the authors write that they assess the bone formation (I did not find the information on bone formation in the results/discussion), as well as OC/CTX ratio is not described. No clear information on bone turnover is presented neither in Results, no Discussion sections.

Regarding your comment about bone formation, we assessed bone formation by measuring osteocalcin (OC), which is a well-established marker of bone formation. We also measured the resorption marker, C-terminal telopeptide of type I collagen (CTX-I), to asses bone resorption. Additionally, we calculated the ratio of OC/CTX-I to assess the rate of bone turnover. In our study, we observed that vegetarian children had higher serum concentrations of CTX-I than omnivores, while the levels of OC were comparable. We have mentioned this in the Results section of our manuscript. Additionally, we determined that the ratio of OC/CTX-I was slightly lower in vegetarian children compared with omnivores, suggesting no significant difference in the rate of bone turnover. We have included this in the Discussion section of our manuscript. We hope that this explanation clarifies the results presented in our study.

Line 426 : what are the advantage of other bone markers ? What information will they permit to obtain ? Please make it clear for the reader.

There are several advantages to using other bone markers in addition to the ones (OC, CTX-I) we measured in our study. The balance between bone formation and resorption processes is controlled by many factors that influence osteoblast and osteoclast activity and are regulated through signaling pathways, such as the complex of RANK/RANKL/OPG cytokines. An elevated ratio of RANKL/OPG is associated with an imbalance in osteoclastic/osteoblastic activity, leading to increased bone resorption and decreased bone mass.

Also, the Wnt-beta-catenin signaling pathway plays an important role in bone metabolism. Its activation stimulates the bone formation process by increasing osteoblast proliferation and differentiation. Its regulation is dependent on many proteins, including sclerostin and Dickkopf-related protein-1 (Dkk-1). Sclerostin and Dkk-1 are antagonists of canonical Wnt signaling, which after binding to the receptor, block the signaling cascade leading to inhibiting bone formation. Sclerostin, synthesized by osteocytes, decreases bone formation by inhibiting the differentiation of osteoblasts and by promoting their apoptosis. Dkk-1 is involved in the mechanism promoting the differentiation of mesenchymal stem cells into adipocytes over osteoblasts. Thus, the Wnt-beta-catenin pathway plays an important role in the regulation of osteoclastogenesis and bone mass, as well as osteocyte and adipocyte function.

Since in the present study, some significant correlations are observed between concentrations of AAs and bone metabolism markers (especially OPG) in vegetarians, it is worth evaluating the regulatory pathway of RANK/RANKL/OPG and, in addition, the Wnt pathway with sclerostin and Dkk-1.

Lines 428-439 : the Conclusion is not clear.

We changed the conclusion to be more clear.

Reviewer 2 Report

I commend the Authors for their nice work. The manuscript is overall well written and clearly presented and the topic is of interest. I just have few minor comments for the Authors:

- do you have any data on other metabolic markers such as lipid profile, serum uric acid, serum urea and serum creatinine? Also, are you planning to analyse urine? It would be interesting to understand if children with less amino acids introit have compensatory renal mechanisms. In fact, kidney plays an important role in amino acids metabolism (PMID: 14749222) and plasma and urine amino acids have shown to differ significantly in children and adolescents with/without diabetes (and the consequent renal hyperfiltration) (PMID: 35507146, PMID: 35523653). Diabetes induces a macroscopic change in tissue metabolism that may represent a good model to understand metabolic regulation in different nutritional states. 

- as vegetarian diets have shown to reduce cardiovascular disease burden in adults, it would be interesting to know if participants of your study already display different metabolic and cardiovascular profile. I suggest the Authors to consider the use of arterial applanation tonometry, the gold standard technique to assess arterial stiffness, to investigate changes in vascular profile of children grown up with vegetarian, vegan or omnivorous diets. 

- another important player in the impact of diet and amino acids availability is the intestinal microbiota (PMID: 29902437, PMID: 29756574). Furthermore, the latter is associated with vascular ageing (PMID: 35743626) so it may mediate the protective effects of vegetarian diet on the cardiovascular system. I suggest the Authors to include this possible pathway of diet-induced amino acids pool regulation. It would be of great interest studying the microbiome of the participants.

Author Response

Comments and Suggestions for Authors

I commend the Authors for their nice work. The manuscript is overall well written and clearly presented and the topic is of interest. I just have few minor comments for the Authors:

- do you have any data on other metabolic markers such as lipid profile, serum uric acid, serum urea and serum creatinine? Also, are you planning to analyse urine? It would be interesting to understand if children with less amino acids introit have compensatory renal mechanisms. In fact, kidney plays an important role in amino acids metabolism (PMID: 14749222) and plasma and urine amino acids have shown to differ significantly in children and adolescents with/without diabetes (and the consequent renal hyperfiltration) (PMID: 35507146, PMID: 35523653). Diabetes induces a macroscopic change in tissue metabolism that may represent a good model to understand metabolic regulation in different nutritional states. 

Thank you very much for your valuable review of our manuscript. Thank you also for the suggested references.

We previously published data on lipid profile parameters, serum levels of adipokines (leptin, adiponectin, resistin, visfatin, vaspin, and omentin), as well as myokine (myostatin, irisin) concentrations in children following vegetarian diets:

  1. Laskowska-Klita T et al. The leptin and some lipid parameter levels in children on a vegetarian diet. Pol Merkur Lekarski 2004, 16(94):340-343.
  2. Ambroszkiewicz et al. Serum concentration of lipids and adipocytokines in prepubertal vegetarian and omnivorous children. Med Wieku Rozwoj 2011, 15(3):326-334.
  3. Ambroszkiewicz et al. Low serum leptin concentration in vegetarian children. Ann Acad Med Bialost 2004, 49:103-105. 
  4. Ambroszkiewicz et al.  Anti-inflammatory and pro-inflammatory adipokine profiles in children on vegetarian and omnivorous diets. Nutrients 2018, 10(9):E1241.
  5. Ambroszkiewicz et al. Bone status and adipokine levels in children on vegetarian and omnivorous diets. Clin Nutr 2019, 38(2):730-737.
  6. Ambroszkiewicz et al. Does a vegetarian diet affect the levels of myokine and adipokine in prepubertal children? J Clin Med 2021, 10: E3995.

So far, we did not analyze serum uric acid, urea, creatinine, or urine amino acid levels in vegetarians. Currently, we have conducted a project on the adipokine and myokine panel in children following different dietary patterns.

We would like to thank the reviewer for the suggestions regarding the role of the kidney in amino acid metabolism. We have read the suggested papers (PMID: 14749222, PMID: 35507146, and PMID: 35523653). Thank you for the suggestions regarding diabetes, which may represent a good model for understanding metabolic regulation in different nutritional states. Although, these issue was not a direct goal of the present manuscript, they might be an inspiration for future research.

- as vegetarian diets have shown to reduce cardiovascular disease burden in adults, it would be interesting to know if participants of your study already display different metabolic and cardiovascular profile. I suggest the Authors to consider the use of arterial applanation tonometry, the gold standard technique to assess arterial stiffness, to investigate changes in vascular profile of children grown up with vegetarian, vegan or omnivorous diets. 

We would like to thank the reviewer for the above suggestions.

 - another important player in the impact of diet and amino acids availability is the intestinal microbiota (PMID: 29902437, PMID: 29756574). Furthermore, the latter is associated with vascular ageing (PMID: 35743626) so it may mediate the protective effects of vegetarian diet on the cardiovascular system. I suggest the Authors to include this possible pathway of diet-induced amino acids pool regulation. It would be of great interest studying the microbiome of the participants.

Thank you for your remark regarding the impact of diet and amino acids availability on the intestinal microbiota. Although it was not the focus of this study, we have reviewed the suggested papers with interest (PMID: 29902437, PMID: 29756574, and PMID: 35743626). We hope that in the future, we will have the opportunity to study the microbiome of children on vegetarian and omnivorous diets.

Additionally, we rewritten the Discussion section according to reviewer 1.

Reviewer 3 Report

The manuscript by Ambroszkiewicz et al „Dietary Intake and Circulating Amino Acid Concentrations in Relation with Bone Metabolism Markers in Children Following Vegetarian and Omnivorous Diets” is well written and provides interesting information.

There are some small points I would like to mention, where I wonder consideration could improve the manuscript.

Fasted blood samples were used, which seem suitable to describe the AA status, but I imagine most of the day children are in postprandial state with different (higher) amino acid levels. Clearly this cannot be measured, but is it a principle limitation?

Could a more complex statistical evaluation using multiple linear regression provide a more detailed insight?

Was the sample size somehow determined in advance? Please describe corresponding considerations in the manuscript.

Would it be helpful to consider AA intake per kg body weight as a parameter for associations with AA status or bone markers?

Specific points

Line 105: what was used as reference for the z-scores?

Line 138: it should be short term dietary influence (influence of recent meal)?

Line 244: It is not clear how “Partial Spearman correlation” was determined?

Line 283: delete “a”

Author Response

Comments and Suggestions for Authors

The manuscript by Ambroszkiewicz et al „Dietary Intake and Circulating Amino Acid Concentrations in Relation with Bone Metabolism Markers in Children Following Vegetarian and Omnivorous Diets” is well written and provides interesting information.

There are some small points I would like to mention, where I wonder consideration could improve the manuscript.

Thank you very much for reviewing our work.

Fasted blood samples were used, which seem suitable to describe the AA status, but I imagine most of the day children are in postprandial state with different (higher) amino acid levels. Clearly this cannot be measured, but is it a principle limitation?

Thank you for this remark. While it is true that most of the day, children are likely in a postprandial state with higher amino acid levels, using fasted blood samples is a common method in research to assess the amino acid status in both children and adults. It is not possible to measure amino acid levels throughout the day in a practical research setting. We believe that our study provides valuable insights into the amino acid status of children following a vegetarian diet.

Could a more complex statistical evaluation using multiple linear regression provide a more detailed insight?

We added a new Table 5, which shows the results of multiple linear regression estimation with osteoprotegerin (OPG) as the dependent variable. Six significantly correlating amino acids and albumin were included as independent variables. Three factors entered the final model in order: alanine, ornithine, and aspartate. The model explains 35.1% of the variation in OPG in the vegetarian group, and the tolerance and VIF indices are highly favorable.

Was the sample size somehow determined in advance? Please describe corresponding considerations in the manuscript.

As mentioned in the Materials and Methods section of the manuscript, the maximum possible number of children consuming since birth was recruited. We have included information about small sample size in the limitation section of our manuscript.

In Poland, vegetarianism in young children is not very common, but in vegetarian families, children often consume a similar diet to their parents. Currently, our institute provides nutritional care for around 110 children (age <18 years old) following vegetarian diets. We plan to include larger groups of vegetarian children in future studies.

Would it be helpful to consider AA intake per kg body weight as a parameter for associations with AA status or bone markers?

We think that recalculating the dietary AAs intake per kg body weight in healthy children would not have much relevance from the research scope of the current work. In the present paper, we analyzed correlations between bone metabolism markers and serum amino acid concentrations. However, we realize that such conversion is important for patients with metabolic diseases involving disorders of amino acid metabolism, e.g. phenylketonuria, where the dietary intake of phenylalanine is adjusted according to the serum levels found.

Specific points

Line 105: what was used as reference for the z-scores?

We added (in Materials and Methods) references for BMI z-scores [KuÅ‚aga et al. Percentile charts for growth and nutritional status assessment in Polish children and adolescents from birth to 18 years of age. Standardy Medyczne 2015, 12, 119–135].

Line 138: it should be short term dietary influence (influence of recent meal)?

We have added the phrase “short-term” in the Materials and Methods section.

Line 244: It is not clear how “Partial Spearman correlation” was determined?

We corrected these sentences. We performed correlation analysis using the non-parametric Spearman test.

Line 283: delete “a”

We corrected this.

Additionally, we rewritten the Discussion section according to reviewer 1.

Round 2

Reviewer 1 Report

The authors have significantly improved the discussion section, the results are clear  and the conclusion is supported by the results. I suggest the accepting of the manuscript in the present form.

Reviewer 3 Report

Thank you very much for considering/discussing the points indicated